# Use of Milk Infrared Spectral Data as Environmental Covariates in Genomic Prediction Models for Production Traits in Canadian Holstein

**DOI:** 10.3390/ani12091189

**Published:** 2022-05-06

**Authors:** Francesco Tiezzi, Allison Fleming, Francesca Malchiodi

**Affiliations:** 1Department of Agriculture, Food, Environment and Forestry, University of Florence, 50144 Firenze, Italy; 2Department of Animal Science, North Carolina State University, Raleigh, NC 27695, USA; 3Lactanet Canada, Guelph, ON N1K 1E5, Canada; afleming@lactanet.ca; 4The Semex Alliance, Guelph, ON N1H 6J2, Canada; fmalchiodi@semex.com

**Keywords:** genomic selection, spectral data, genotype by environment interaction

## Abstract

**Simple Summary:**

Genomic selection models aim at predicting the performance of individuals with the use of genomic markers. In animal breeding, prediction models are seldomly tested for their ability to predict new individuals’ performance under different environmental conditions, despite the changes in management and diet that the industry undergoes. In this study, we propose a method to use milk infrared spectra as descriptors of environmental variation among herds. These descriptors can be incorporated in genomic prediction models similarly to how genomic markers are included. The inclusion of environmental descriptors is shown to improve the predictive ability for new genotypes under new environmental conditions.

**Abstract:**

The purpose of this study was to provide a procedure for the inclusion of milk spectral information into genomic prediction models. Spectral data were considered a set of covariates, in addition to genomic covariates. Milk yield and somatic cell score were used as traits to investigate. A cross-validation was employed, making a distinction for predicting new individuals’ performance under known environments, known individuals’ performance under new environments, and new individuals’ performance under new environments. We found an advantage of including spectral data as environmental covariates when the genomic predictions had to be extrapolated to new environments. This was valid for both observed and, even more, unobserved families (genotypes). Overall, prediction accuracy was larger for milk yield than somatic cell score. Fourier-transformed infrared spectral data can be used as a source of information for the calculation of the ‘environmental coordinates’ of a given farm in a given time, extrapolating predictions to new environments. This procedure could serve as an example of integration of genomic and phenomic data. This could help using spectral data for traits that present poor predictability at the phenotypic level, such as disease incidence and behavior traits. The strength of the model is the ability to couple genomic with high-throughput phenomic information.

## 1. Introduction

Genomic prediction models are oriented towards predicting the performance of new individuals with the use of genomic markers. The ability of genomic prediction models to predict across environments in livestock is seldomly tested. In addition, these environmental effects will exert the same pressure on each candidate; such models do not need to account for effects other than the genetic one. However, environmental components definitely exert an effect on the phenotype, which needs to be considered in extrapolating predictions to new environments, i.e., ensuring the portability of the predictions to different environmental conditions. In addition, these environmental changes could exert different effects depending on the genotype, and different genotypes may react differently to environmental changes [1,2]. In other words, there could be a presence of genotype by environment interaction (GxE).

In breeding plans, the new genotypes are usually tested in herds (or flocks, or stations) that are part of an established organization, which means that these genotypes are tested under known environmental conditions. Sometimes, new herds could join the breeding program, or the same herds could make changes in management, either intentionally or unintentionally [3,4]. In addition, some environmental conditions are not fully controllable by the farmers, meaning that some environmental variables could fall outside the known range. For example, climate change could lead livestock to experience environmental conditions not experienced before [5,6], making the heat load parameter fall outside of the known range. Figure 1 shows the three different scenarios for the prediction of performance under new genotypes and/or environmental conditions.

Here, sires (i.e., genotypes, families) are reported in rows, while herds (i.e., environments) are reported in columns. Each herd show several herd-year-season classes, which define the temporary environmental variance within the permanent location. Sires 1 to 10 are considered known or proven. Likewise, herds 1 to 4 are considered known environments, already being part of the breeding organization. Phenotypic information belonging to sires 1 to 10 and herds 1 to 4 is considered a training set and labeled as section ‘A’. The performance of known sires could be extrapolated to new herds (5 and 6), which differ from the known ones for a number of parameters. This would correspond to section B, i.e., known genotypes in new environments. Conversely, the prediction of new sires’ performance (11 to 16) in known herds is what is commonly practiced in genomic selection, and that would correspond to section C. Finally, the most challenging scenario would be to predict the performance of new sires in new herds, i.e., section D. Although scenarios C and D might seem unrealistic, we should keep in mind that herd management and environmental conditions are in constant change, and this scenario is more relevant when farming conditions (e.g., diet, climate) tend to change quickly. Therefore, there is aneed to consider predictions for new environments in the same way that we consider predictions for new genotypes. For the former, the use of environmental covariates might be useful.

In the implementation of cross-environment genomic prediction, the choice of the environmental covariates is not trivial. Several attempts have been taken in using different sets of variables, from climate records to management parameters [7,8,9], and from geographical to spatial coordinates [10,11]. Some authors have also proposed the use of the estimates for contemporary group itself or some a posteriori estimation of it, in order to make full use of the data [12,13]. Here, a first model needs to be implemented using the contemporary group as a fixed, cross-classified effect. Best linear unbiased estimates (BLUE) of contemporary groups are then merged back to the original dataset and used as a fixed linear covariate. This is needed to obtain variables that only contain environmental variability and are not collinear to the genetic component (or any other). The average phenotypic value of a contemporary group could include some environmental variation, because not all genotypes are included in every contemporary group. Therefore, such BLUE provide an average contemporary group performance, adjusted for genetic (and other) effects; therefore, their variation is determined by the environmental component alone. However, this method shows a pitfall in cross-validation, since the trait itself needs to be recorded in the environmental classes (i.e., herds) used for validation. This brings the need to use a predictor that can be (easily and cheaply) recorded in new environmental classes before the actual trait has to.

In the era of high-throughput phenotyping, there is a need to re-think animal breeding models to account for the vast amount of data generated by the fast-developing field of sensor technology [14]. Fourier-transformed, mid-infrared spectral (FTIR) information offers an alternative as an inexpensive, high-throughput predictor or indicator. FTIR is largely used in the agricultural and livestock industries for the high-throughput assessment of several qualitative measures, especially when the phenotype would be expensive to measure using wet-lab chemistry. Several studies have investigated how to include this source of information into breeding programs [15,16,17]. One potential use of spectral information could be to include all the spectral variables into the model as environmental descriptors [18]. This approach does not require the development of calibration equations for the prediction of specific variables; instead, it only requires the extraction of the environmental component from this variable (and the removal of the genetic component). The calibration equation used in this approach will be implicit in the use of the spectral covariates in the prediction model, which can be updated at every round of genetic evaluation. Even in absence of knowledge about the association between specific wavenumbers and the trait of interest, this allows us to inform the models about the ‘environmental coordinates’ of the phenotype to predict. If, supposedly, some wavenumbers are associated with the presence of certain fatty acids in the milk, and it is the herd diet determining the presence of those fatty acids, the FTIR absorbance at those wavenumbers will contain information about the diet. In support of this hypothesis, FTIR spectral data have been used to successfully discriminate milk samples based on feeding or grazing systems [19,20]. The spectral wavelengths have also been shown to be in association with diseases incidence [21] and the cows’ metabolic status [22], which suggest the strong relationship between the spectral data and the herd status, in general. Overall, expecting the presence of systematic environmental variation in the milk spectrum, such environmental variation could be used for different modeling purposes at no cost because of the routine collection and storage of spectral data. The environmental variation present in the spectrum could be associated with certain environmental conditions (e.g., heat stress), therefore giving the possibility to be used as indicator. In addition, the spectral data show high dimensionality, which relates to the possibility to capture complex environmental variation. All this could be exploited in prediction models aimed at predicting outside of the known environmental range, but a proper (cross-) validation is needed.

The objective of this study was to test the value of FTIR information as environmental covariates in genomic prediction models, for the prediction of new genotypes in new environments.

## 2. Materials and Methods

### 2.1. Phenotypic Data

Phenotypic data used in this study came from the Lactanet Canada database. Test-day records for milk yield (MY), somatic cell score (SCS), and mid-infrared spectral data were made available together with test dates, herd identifiers, and pedigree information. Spectra data included absorbance for 1060 wavenumbers (WVN) from FOSS MilkoScan FT6000 spectrophotometers (Foss, Hillerød, Denmark). Each WVN refers to a specific wavelength in the infra-red range of the spectrum [23]; the WVN have a discontinuous numbering from 5010 to 925 cm^−1^. Information about lactation number and stage of lactation was also available. The herd-year-season class (HYS) was created by combining year and season of test date with the herd. Seasons were created as in Rovere et al. [23], i.e., January–March, April–June, July–September, and October–December. Stage of lactation classes were created as 13 monthly classes, with the 13th class including records up to the 18th month of lactation. An open ‘fourth or later’ lactation number class was also created. Stage and number of lactations were combined into 52 number by stage of lactation (NSL) classes.

The initial dataset included more than 10 million records on 1.3 million cows. Editing was performed by removing test-day records beyond the 18th month of lactation (540 days), removing cows with less than 12 records available and HYS classes with less than 20 records. Merging the spectral data generated a dataset that included 1,540,935 records, from 84,131 cows. Records were removed if records for any of the spectral variables exceeded the five standard deviations from the mean. Hereinafter, this dataset will be referred to as Data1_full_. A reduced dataset, including only herds with at least 30 HYS classes, was also created and will be referred to as Data1_reduced_ (N = 571,440). Both datasets are described in Table 1.

### 2.2. Genomic Data

Genomic information for 1992 bulls was provided by The Semex Alliance. Individuals were genotyped with the Illumina SNP50 Beadchip. A total of 45,187 SNPs were available. Editing included the removal of markers with call rate below 0.90 and minor allele frequency below 0.05. After editing, 38,024 markers were left for analysis. Only 483 bulls will be then used for analysis; see below.

#### 2.2.1. Data Analysis Overview

This study proposes a multi-step approach. First, environmental information had to be extracted from the phenotypic and spectral data (steps 1 and 2). Then, the Kernel-based genomic prediction models were implemented on the condensed information as extracted from the first step. The Kernel-based models were used for variance components estimation (step 3) and cross-validation (step 4).

#### 2.2.2. Step 1: Estimation of Herd-Year-Season Deviations

Datasets Data1_full_ and Data1_reduced_ were used in the first step of the study. This step aimed at the estimation of Best Linear Unbiased Estimates (BLUEs) for the HYS classes in order to build environmental covariates to be used in the next steps [12,13]. Milk yield, SCS and each spectral wavenumber was analyzed using the following model:(1)$$y_{\mathrm{ijkl}}={\mathrm{NSL}}_{i}+{\mathrm{HYS}}_{j}+a_{k}+p_{k}+\epsilon_{\mathrm{ijkl}}$$
where *y_ijkl_* is the phenotypic record for the l^th^ observation of the k^th^ cow, in the j^th^ HYS and ith NSL class, *NSL_i_* is the fixed effect of the i^th^ number by stage of lactation class, *HYS_j_* is the fixed effect of the j^th^ herd-year-season class, *a_k_* is the additive genetic effect of the k^th^ individual, *p_k_* is the cow permanent environmental effect of the k^th^ individual, and *ϵ_ijkl_* is the residual.

The vector of additive genetic effects was defined as $a\;\sim \;N\left(0,\;A\sigma_{a}^{2}\right)$, where A is a pedigree-derived numerator relationship matrix and $\sigma_{a}^{2}$ is the additive genetic variance. The vector of cow permanent environmental effects was defined as $p\;\sim \;N\left(0,\;I\sigma_{p}^{2}\right)$, where I is an identity matrix and $\sigma_{p}^{2}$ is the cow permanent environmental variance. The vector of residuals was defined as $\epsilon \;\sim \;N\left(0,\;I\sigma_{\epsilon }^{2}\right)$, where $\sigma_{\epsilon }^{2}$ is the residual variance and I is an identity matrix.

Variance components estimates for each wavenumber and production traits were obtained using Data1_reduced_ using the software Gibbs2f90 [24] version 1.86; a total of 35,000 iterations were run, discarding the first 5000 as burn-in and thinning every 10 iterations. As no specifications about the prior were passed to the program, the variance components were assigned flat priors (i.e., no shrinkage) and fixed effects were assigned bounded uniform priors. For the random effects, the software uses normal distribution priors with mean equal to 0 and variance equal to the estimated variance at each iteration. Convergence was assessed by visual inspection of trace plots and Geweke’s test implemented in the ‘coda’ package in R [25]. Once the variance components were estimated, the model solutions and their standard errors (for the fixed and random effects) necessary for the following steps were obtained using Data1_full_ and the software BLUPf90 [26] by fixing the variance components to the estimated values.

Heritability ($h^{2}$) and repeatability ($r^{2}$) were defined as $h^{2}=\frac{\sigma_{a}^{2}}{\sigma_{a}^{2}+\sigma_{p}^{2}+\sigma_{e}^{2}}$ and $r^{2}=\frac{\sigma_{a}^{2}+\sigma_{p}^{2}}{\sigma_{a}^{2}+\sigma_{p}^{2}+\sigma_{e}^{2}}$.

A dataset containing all the BLUEs for the HYS effect was created, including solutions for MY, SCS, and all the 1060 wavenumbers. The HYS BLUEs for each MY, SCS, and WVN were centered to null mean and unit variance. Principal components (based on the covariance matrix) were extracted from the HYS BLUEs for the 1060 wavenumbers using the native R function *princomp* [27].

#### 2.2.3. Step 2: Calculation of Herd-Year-Season Daughter-Yield-Deviations

Using solutions from models in step 1, daughter-yield deviations for each bull in each herd-year-season class (hysDYD) were calculated [9]. The hysDYD were defined by pre-correcting the phenotypic data for the number by stage of lactation class, then averaged by sire-HYS class. The hysDYD were then weighted for the within-HYS effective daughter contribution (EDC) to account for the different amount of information contained in each hysDYD value, given by the different number of sire’s daughters and their records. Here, EDC were calculated following the sixth method proposed by Fikse and Banos [28]. Using Data1_full_, a total of 518,669 hysDYD observations were made available. These were edited by keeping only those belonging to genotyped sires and showing an EDC of at least 1.6, forcing sires to be present in at least 3 HYS classes, HYS classes to show at least 3 sires and herds having at least 3 HYS classes. This generated Data_2_, which contained 16,891 observations on 483 sires, 406 herds and 3316 HYS classes (Table 2).

#### 2.2.4. Step 3: Estimation of Variance Components

Different models including additive genetic and environmental effects were implemented, as well as models that included their interaction. Variance components estimation was carried out using dataset Data_2_. A model with the two main effects was first defined as:(2)$$y_{\mathrm{ij}}=\mu +g_{i}+e_{j}+\epsilon_{\mathrm{ij}}$$
where *μ* is the intercept, *y_ij_* is the hysDYD for the i^th^ sire in the j^th^ HYS, *g_i_* is the additive genetic effect of the i^th^ sire, *e_j_* is the environmental effect for the j^th^ HYS, and *ϵ_ij_* is the random residual. The vector of additive genetic effects was defined as $g\;\sim \;N(0,G\sigma_{g}^{2}$), where G is a genomic relationship matrix and $\sigma_{g}^{2}$ is the estimated additive genetic variance. The genomic relationship matrix was built according to the first method described by VanRaden [29] using the software preGSf90 [30]. The vector of environmental effects e_j_ was defined in different ways depending on the model. A baseline model (Base) was defined and considered the environmental effect as random uncorrelated, i.e., $e\;\sim \;N(0,\;I\sigma_{e}^{2}$), where I is an identity matrix and $\sigma_{e}^{2}$ is the environmental variance. A set of models considered the environmental effect which included covariance matrices among the HYS classes (also known as Kernels [31,32]). Here, the vector of environmental effects was defined as $e\;\sim \;N(0,\;E\sigma_{e}^{2}$), where E is a matrix that reports the covariance among the HYS classes based on the environmental covariates. Such covariance matrix is built as E=XX′, where X is a matrix that contains the environmental covariates (in columns) for each HYS class (in rows). The columns in X report the environmental covariates (e.g., BLUE for MY, SCS and the WVN) centered to null mean and unit variance, so that the E matrix does not need to be rescaled.

In fitting the model, since data involve multiple records per sire and HYS class, the genetic covariance structure of additive effects was defined as ZGZ′, while the environmental covariance structure was defined as WEW′, where Z is the (sire by observation) incidence matrix and W is the (HYS class by observation) incidence matrix [7], while G and **E** were as defined above.

The choice of the environmental covariates to be included in **X** defined the model. In model AWN, HYS BLUEs for all the 1060 wavenumbers were used. This means that **X** was a matrix with 1060 columns and a number of rows equal to the number of HYS included in the analysis, i.e., 16,891 (Table 2). In models PC2, E is a two-column matrix with the first two principal components extracted from the HYS BLUEs. Similarly, PC10 included the first 10 principal components, while model PC20 included the first 20, PC30 included the first 30 and model PC40 included the first 40 principal components. The first 2, 10, 20, 30 and 40 components absorbed 62.7%, 89.5%, 93.8%, 95.8%, and 96.6% of the whole WVN variance, respectively. In model PROD, covariance was defined on HYS BLUEs for MY and SCS, defining a X matrix with 2 columns.

A summary of the environmental covariates used in the models is reported in Table 3.

A model with the interaction between the additive genetic and environmental effects was also defined, as in:(3)$$Y_{\mathrm{ij}}=\mu +g_{i}+e_{j}+{\mathrm{ge}}_{\mathrm{ij}}+\epsilon_{\mathrm{ij}}$$
where *y_ij_*, *μ*, *g_i_*, *e_j_* and *ϵ_ij_* are as defined in equation [3], and *ge_ij_* is the interaction term. The vector for the interaction effect was defined as $\mathrm{ge}\;\sim \;N(0,\;\left[ZGZ\prime {}^{\circ}WEW\prime \right]\sigma_{\mathrm{ge}}^{2}$), where ZGZ′°WEW′ denotes the Hadamard product between the additive genetic and environmental kernels, and $\sigma_{\mathrm{ge}}^{2}$ is the variance for the interaction term [7,9]. Again, the environmental component was defined in different ways depending on the environmental covariates included in E. For each set of environmental covariates, a model as in Formula (2) (without interaction term) and a model as in Formula (3) (with the interaction term) were implemented.

#### 2.2.5. Step 4: Cross-Validation

The relevance of each set of environmental covariates to improve the prediction models was tested using cross-validation. As outlined above, the predictive ability for models that differ for their inclusion of environmental covariates needs to be tested on schemes that introduce new genotypes but also new environments.

The model validation was carried out as a repeated four-fold cross-validation. This involved the following steps: (1) sampling 100 sires and 100 herds from Data_2_; (2) defining the validation sets (B, C, and D altogether, Figure 1) as the records belonging to the sampled 100 sires and 100 herds; (3) defining the training set (A, as pictured in Figure 1) as the remainder of Data_2_; (4) fitting the models on the training set in order to obtain solutions for all the sires and HYS classes, including those for which the phenotypes were removed from the training set; (5) obtaining predictions for the training and validation sets by summing the respective solutions as defined by each model; (6) calculating the prediction accuracy as the Pearson correlation between predicted and observed hysDYD. The prediction accuracy was calculated separately for each of the sections B, C, and D: the hysDYD that belonged to the sampled herds and the non-sampled sires were assigned the B validation set, the hysDYD that belonged to the sampled sires and the non-sampled herds were assigned the C validation set, and the hysDYD that belonged to both the sampled sires and sampled herds were assigned to the D validation set.

The six steps of the cross-validation were repeated 20 times in order to have an appropriate representation of sires and herds in the training and validation sets. For each of the 20 replicates, a unique combination of new sires and new herds was sampled to define the four sections (A, B, C, and D), and each replicate generated a value of accuracy for each section. On average, 60% of the records were assigned to the training set (section A), 40% of the records were assigned to the validation set, 18% in section B, 15% in section C and 7% in section D. The mean and standard deviation of the accuracy over the 20 replicates were calculated and used to compare model performance.

#### 2.2.6. Model Implementation for Steps 3 and 4

All models were fitted using the R function BGLR [33]; kernels were used using the Reproducing Kernel Hilbert Spaces definition after eigenvalue decomposition was carried out [34]. The environmental effect in Base was implemented as a Bayesian Ridge regression. A total of 75,000 iterations were run, discarding the first 25,000 as burn-in and thinning every 10 iterations. Convergence was assessed by visual inspection of trace plots and the Geweke’s test implemented in the ‘coda’ package in R [25].

## 3. Results

### 3.1. Variance Decomposition of Spectral Data

Heritability and repeatability for each of the 1060 WVN is reported in Figure 2. Estimates of heritability and repeatability for MY and SCS are also reported as horizontal lines in the same plot (solid line for MY, dashed line for SCS). Estimates of heritability were 0.20 and 0.11 for MY and SCS, respectively, while estimates of repeatability were 0.40 and 0.34.

The heritability estimates show different values across the WVN range, and repeatability estimates appear to follow a pattern similar to the heritability, suggesting that the cow permanent environmental contribution is constant across the wavenumbers. Both heritability and repeatability were stable between, approximately, WVN 5000 and 4200 (with values around 0.18 and 0.23, respectively); a decline in both parameters’ estimates brought them to the null value around WVN 3600, with the exception of a peak around WVN 3700 (0.30 for heritability, 0.38 for repeatability). The null estimate values were found for WVN from 3600 to 3070, with the exception of a short peak around WVN 3450 (0.03 for heritability, 0.04 for repeatability). Starting from WVN 3050, both parameters increased dramatically for most of the remainder of the WVN, with some exceptions. While the heritability values were, on average, around 0.35, and the repeatability values were around 0.45, lower values were found between WVN 2600 and 2500 (0.2 heritability, 0.25 repeatability) and between WVN 1670 and 1610, showing null values for both parameters. Below WVN 1610, both parameters showed irregular estimates, with heritability being 0.38 and repeatability being 0.46, on average.

### 3.2. Genotype, Environment and Their Interaction on the Studied Traits

Figure 3 and Figure 4, on the top panel, the proportion of variance explained by the additive genetic effect (G), environmental effect (E), and their interaction (GxE) term. For MY, the E component was the strongest effect, at least for models Base and PROD. When spectral kernels were used, the E component was still large with model AWN but smaller with models PC10 to PC40. The E component was almost null with model PC2. The G component was somewhat constant across models, with a slight inflation when the E component was smaller, suggesting some tradeoff between the two effects. The GxE component was mostly small, with the exception of the PROD and PC models. For SCS, the E component was smaller in magnitude but followed the same pattern, with the exception of the PROD model, which showed low estimates. The G component was constantly larger than the E component. The GxE component was still small but larger in proportion to the other two components, especially for model PROD.

Model Base considered the E component as a random uncorrelated effect; therefore, the solutions for the HYS classes were assumed to be independent and without constraints. This model showed the largest estimates of environmental variance across both traits. For MY, model PROD showed similar E estimates, suggesting that the two covariates (BLUE for MY and SCS) describe the whole variation among herds. For SCS, model PROD showed lower E magnitude than Base, suggesting that there is environmental variation not fully captured by the two covariates. Model AWN was the spectra-enabled model that captured most of the E variance for both traits. The models that used the principal components of the WNV showed scarce ability to absorb variance, although there was an increase in variance absorbed for both traits when increasing the number of principal components used. This suggest that all the variation in the FTIR spectral data could, and should, be used for maximizing the (environmental) variance explained by the model.

The GxE component was small as expected. Still, it absorbed ~4% and ~3% of the variance for MY and SCS, respectively, under model PROD. Other models showed lower estimates of the GxE components.

### 3.3. Cross-Validation, without the Inclusion of the GxE Interaction Term

Results for the three cross-validation scenarios (Figure 1) are reported in the lower panels of Figure 3 and Figure 4 for MY and SCS, respectively. Dots represent the average predictive ability across the twenty replicates; error bars report the standard deviation of the prediction accuracy across the replicates. The black dots (and bars) refer to the models that included the G and E terms, the blue dots (and bars) refer to the models that included the G, E and GxE terms. The model Base included the E term as an uncorrelated random effect (no Kernel included); therefore, the interaction term could not be fitted.

The section B reports a scenario where information from ‘known’ sires is predicted under new environments. Here, the model that used an uncorrelated random effect (Base) to fit the E component underperformed compared to models that included environmental covariates. For MY, the predictive ability for the Base model was 0.20, while models PROD showed 0.75, followed by models AWN with 0.50. The rest of the models showed lower predictive ability, still larger than Base. Only model PC2 showed comparable performance to the Base model. For SCS, again, all covariate-based models performed better than the Base model (0.24), but with smaller margin. The best-performing models were AWN and PROD (0.29 and 0.30, respectively). PC again showed comparable performance to Base.

Section C reports a scenario comparable to genomic selection, with the information from new sires being predicted under known environments. For both traits, only model PROD was able to outperform model Base (0.68 vs. 0.61 for MY, 0.25 vs. 0.16 for SCS). All other models underperformed model Base, with PC2 showing the worst performance.

Section D reports genomic prediction results achieved for new sires into new environments. The two traits showed a different pattern. For MY, all models outperformed Base, which has null (~0.0) prediction accuracy. The best performing model was again PROD, followed by AWN. PC2 was again the lowest performing covariate-informed model (0.10). For SCS, smaller differences were found between models, but some models outperformed Base (0.13). PROD was again the best performing model (0.24), followed by AWN. All other models performed similarly to Base.

### 3.4. Cross-Validation, with the Inclusion of the GxE Interaction Term

All models that included the GxE interaction term performed equally, if not slightly worse, than the respective model without the term. This could be a reflection of the small magnitude of the estimated GxE effect.

## 4. Discussion

### 4.1. Spectral Information in a Precision Livestock Farming Framework

The field of phenomics is rapidly gaining attention, appearing promising for the non-finite nature of the phenomes as opposed, for example, to the genomes [14]. Milk spectral data have already been used for decades yet remain an important source of information at the commercial level. The literature on the use of milk spectral data for predicting phenotypes of interest is vast. Using spectral data, while some traits can be better predicted than others, lately, the interest has shifted towards the integration of spectral and genomic information. In its simplest implementation, spectral-predicted phenotypes can be included in multi-variate models together with the wet-lab measured phenotype of interest [35,36]. This has shown some improvement in predictive ability, although such improvement largely depends on the trait and the quality of the spectral calibration equations. In different implementations, these two sources of data have been successfully integrated in the same models for the prediction of both fat and protein composition [37,38], where both genomic markers and spectral information served as predictors. The difference between the methods proposed in the other studies [37,38], and this study resides in the fact that, here, spectral information was only used for its environmental component. Although the comparison was not made in a straightforward manner, the two approaches serve different purposes. We opted for using only the environmental component of the spectrum for the assumptions that (i) the genomic markers would absorb the genetic component and (ii) the explicit purpose of modeling environmental variation. In addition, the modeling of the GxE component would have been hampered by the presence of genetic variation in the spectral data [12,18].

### 4.2. Variance Decomposition of Spectral Data

This study took advantage of a large dataset that included productive records but also spectral data. To the authors’ best knowledge, this study has used the largest spectral dataset for variance components estimation up to date. The first step in this study involved extracting the environmental variation from the spectral data.

The estimates of heritability for all the spectral wavenumbers show a similar pattern as found in previous research. Both Rovere et al. [23], who worked on a subset of this dataset, and Wang and Bovenhuis [37], who worked on a different dataset, found the same pattern for heritability across the spectral variables.

### 4.3. Spectral Information as Environmental Covariate

Based on results from the present study, the environmental component of mid-infrared spectra could be used as covariate in genomic prediction models. The estimation of variance components for MY and SCS using Kernel regression shows promising results, with a sizable and constant genetic component and different contribution of the environmental kernel depending on the covariates used. Model PROD, which uses the BLUEs of the traits themselves as covariates, shows the largest estimation of variance for MY, but spectral information shows larger estimates for SCS. These results could be due to the fact that SCS shows little HYS variation per se (0.08), which reflects the inability of the MY/SCS BLUE estimates to represent that environmental variation. Conversely, the AWN model was able to capture more environmental variation for SCS.

The approach used in this study for the use of spectral information in genomic prediction models is similar to the one used by Krause et al. [18], who used hyperspectral information (based on reflectance) of wheat canopy to inform prediction models. In that study, an effort to integrate genomic and spectral information into the same model was carried out: the environmental variation for a given genotype in a given site-year was extracted in order to remove the collinearity between the genomic and spectral covariates, which makes it even more similar to this study. Wheat canopy, just like milk composition, is determined by both genetic and environmental effects, so that the same statistical methods can be applied to pursue the improvement of genomic prediction models.

### 4.4. Genomic Predictions across Environments

This study used solely milk yield and somatic cell score as phenotypes of interest. Other routinely recorded traits, such as fat and protein percentage, could not be used because they are nowadays predicted using infrared spectroscopy [39], which would have led to an inherent collinearity between the phenotype and the spectral predictors. Although the relevance of such traits is acknowledged, we could not proceed with the analysis of those traits for this reason.

The results from the cross-validation are different depending on the scenario, as expected. Predictions based on spectral data have been found to provide dramatically different results based on how the training and validation sets were created. Wang and Bovenhuis [37] reported that across-herd predictions of bovine methane emissions showed much lower accuracy than within-herd predictions. Dadousis et al. [40], in predicting goat milk coagulation traits, showed that model predictive ability depended largely on the farm(s) included in the validation set.

In prediction scenario C, to be considered equivalent to a common genomic selection scenario, none of the environmental covariates provided meaningful contribution to the predictive ability of the models. Such covariates appear as non-informative when predictions are drawn to known environments, i.e., there is no need to extrapolate to ‘new’ environmental conditions.

In scenario B, where predictions for the ‘known’ genotypes were extrapolated to new environments, PROD was the best-performing model, with stronger advantage over Base for MY. Unfortunately, model PROD is unrealistic and should only be used for comparison, as an upper bound that the environmental-covariate-informed model could reach. In order to implement model PROD, phenotypes for the ‘new’ environments need to be measured in order to obtain the best linear unbiased estimates for the same environments. If phenotypes are available for these environments, predictions are then not needed, which totally defeats the purpose of predicting the performance in such environments. Spectral information seems promising in providing environmental coordinates for the ‘new’ environments. Model AWN did not outperform model PROD but did outperform model Base in scenarios B and D. Especially in the latter, where predictions for new genotypes are extrapolated to new environments, there was a large advantage of model AWN over model Base, indicating the need to inform the prediction models with environmental covariates but also the opportunity in using the spectral data as sources of environmental variation.

### 4.5. The Dimensionality of Milk Spectral Data as Environmental Descriptors

In this study, we also attempted to reduce the dimensionality of the spectral data by using the most relevant principal components to build the environmental kernel. The use of the principal components did not seem to provide any advantage. Yet, it appears that a number of components larger than 40 would be needed to fulfill the potential of the spectral data to absorb phenotypic variance or predict across environments. In fact, the first 40 principal components only explained about 65% of the total spectral variation, which was made up of 3316 HYS classes.

## 5. Conclusions

The present study showed a simple procedure to include the environmental component of the spectral information into genomic predictions models as a set of covariates using Kernel regression. The results showed that this method was particularly advantageous when genomic predictions for new genotypes under new environmental conditions have to be obtained. Fourier-transformed infrared spectral data represent a useful source of information for the calculation of the ‘environmental coordinates’ of a given farm in a given time.

Farming conditions are evolving, and livestock will be subject to new environmental conditions. Genomic prediction models could take advantage of environmental information in order to extrapolate candidates’ performance to new environments. In general, the goal is to link the different environmental blocks (e.g., herds, herd-year-season classes) using some function that could be reflective of their management strategies or general environmental conditions. The herds are no longer considered independent but are assumed to be connected based on the covariates used. Because of this connection between the herds, a practical implementation of this method could be to obtain FTIR-derived coordinates for each new herd season. These coordinates would then be used in a second genomic prediction model such as the one used in this study. For any trait of interest, the model would yield genomic predictions for new herd seasons based on the genomic information and the FTIR-derived coordinates.

This approach could be particularly advantageous in presence of a large genotype by environment component, which was not detected in this study. Part of this reason could be the limited number of models tested, since the study was oriented towards the use of the spectral data rather than the estimation of this component.

Suggesting a different use of spectral information, this study is an example of the integration of genomic and phenomic data. With the proposed procedure, calibration equations are not needed because only the environmental component of the spectral variables is used. This could help using spectral data for traits that present poor predictability at the phenotypic level, such as disease incidence and behavioral phenotypes. Further research should focus on the reduced computational challenge of incorporating the spectral data.

## Figures and Tables

**Figure 1 animals-12-01189-f001:**
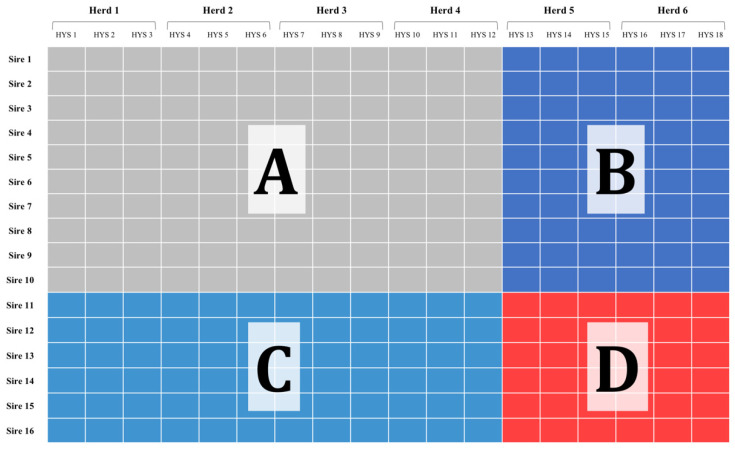
Cross-validation scheme for combinations of genotypes (sires) and environments (herds) to be included in the validation sets. Sires 1 to 10 are considered known or proven. Herds 1 to 4 are considered known environments. Phenotypic information belonging to sires 1 to 10 and herds 1 to 4 could be considered a training set and labeled as section (**A**). The performance of known sires could be extrapolated to new herds (5 and 6); this would correspond to section (**B**), i.e., known genotypes in new environments. Conversely, prediction of new sires’ performance (11 to 16) in known herds corresponds to section (**C**). Finally, the most challenging scenario would be to predict the performance of new sires in new herds, i.e., section (**D**).

**Figure 2 animals-12-01189-f002:**
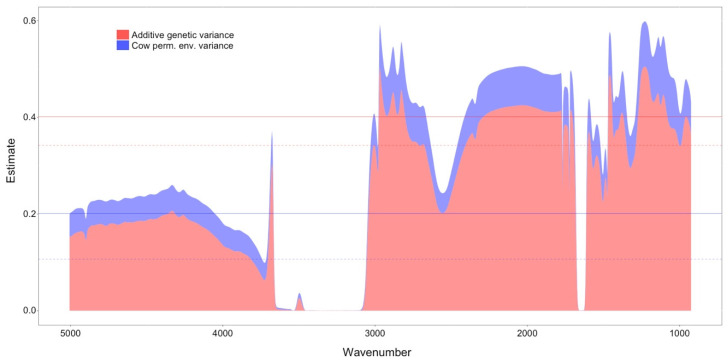
Heritability and repeatability for each of the 1060 wavenumbers used in the study. The solid blue and red lines report the heritability and repeatability estimates for milk yield, while the dashed lines report the same parameter estimates for somatic cell score. Each wavenumber refers to a specific wavelength in the infra-red range of the spectrum and show a discontinuous numbering that goes from 5010 to 925 cm^−1^.

**Figure 3 animals-12-01189-f003:**
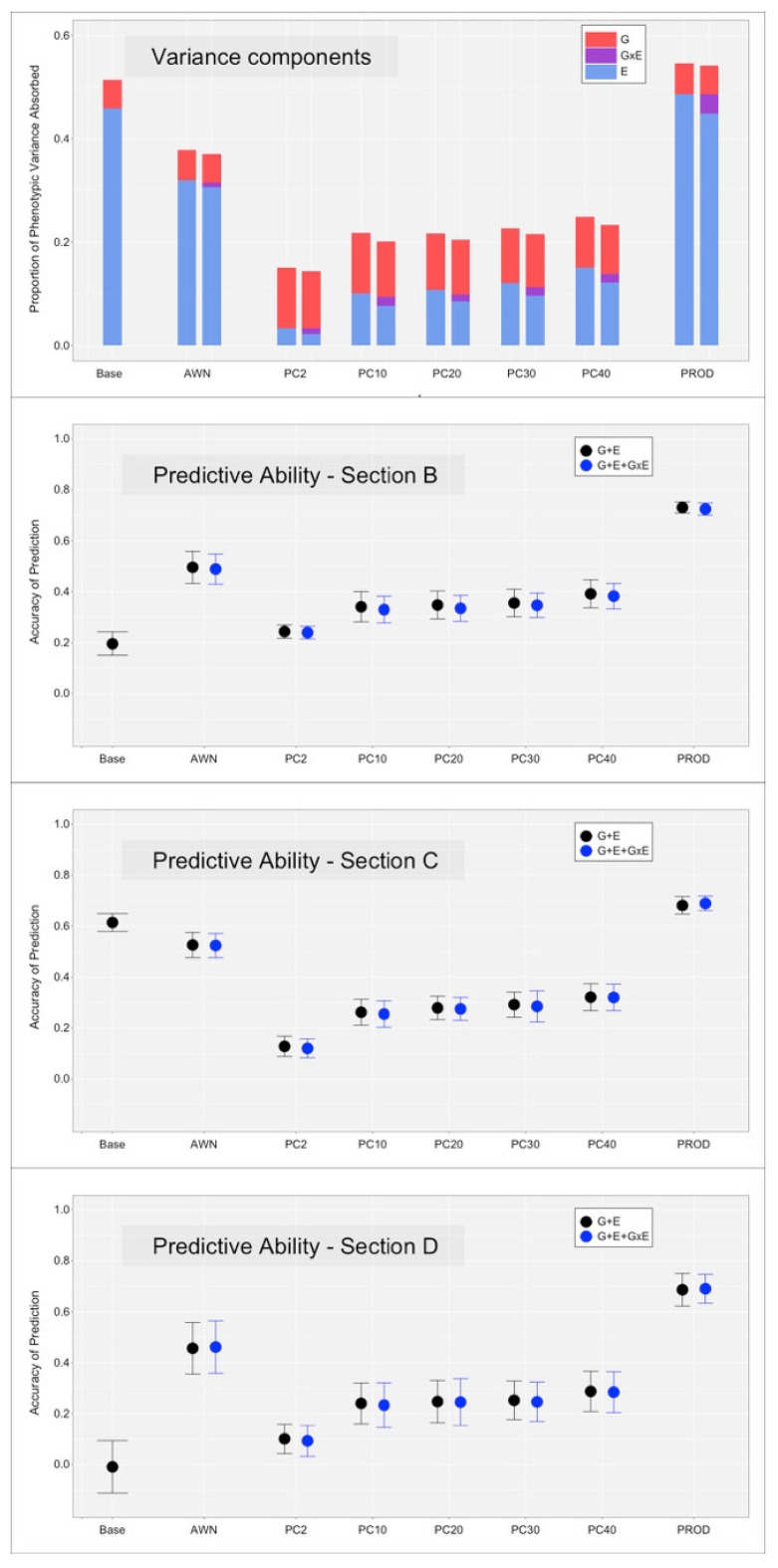
Proportion of variance components explained by the additive genetic effect (G), environmental effect (E) and their interaction (GxE) term, together with predictive ability of the respective models under the different scenarios for milk yield (MY). The black dots (and bars) refer to the models that included the G and E terms; the blue dots (and bars) refer to the models that included the G, E, and GxE terms.

**Figure 4 animals-12-01189-f004:**
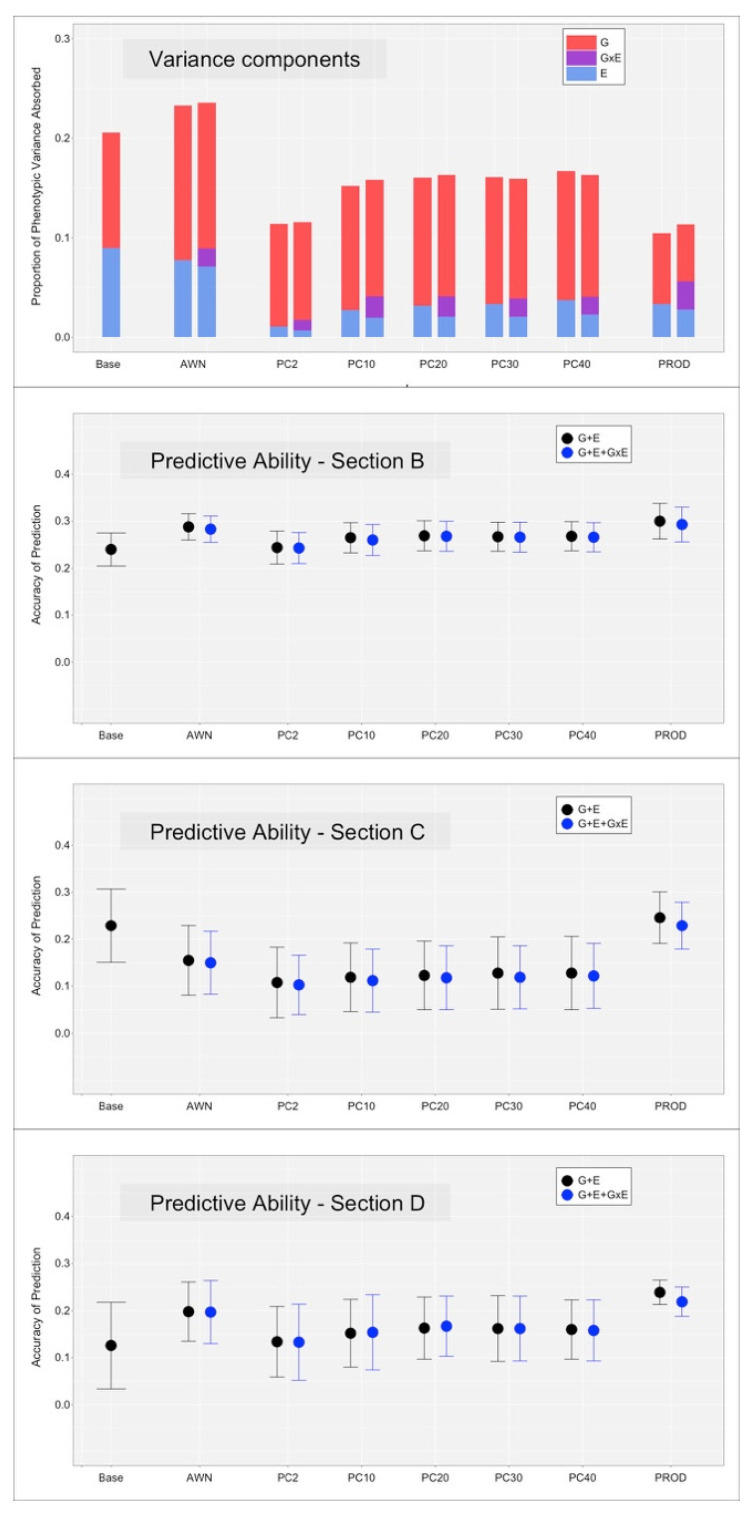
Proportion of variance components explained by the additive genetic effect (G), environmental effect (E), and their interaction (GxE) term, together with predictive ability of the respective models under the different scenarios for somatic cell score (SCS). The black dots (and bars) refer to the models that included the G and E terms, the blue dots (and bars) refer to the models that included the G, E, and GxE terms.

**Table 1 animals-12-01189-t001:** Descriptive statistics for the datasets including test-day records and used in the study.

	Data1_full_	Data1_reduced_
Number of records	1,540,935	571,440
Number of cows	84,131	29,057
Numbers of sires	5759	3540
Number of dams	69,665	23,523
Number of individuals in pedigree	419,586	177,916
Number of herds	768	214
Number of herd-year-season classes	28,222	9766
Milk Yield, kg	35.66 (10.36)	35.47 (10.42)
Somatic Cell Score	2.36 (1.92)	2.38 (1.92)

**Table 2 animals-12-01189-t002:** Descriptive statistics for the datasets including herd-year-season daughter-yield-deviations.

	Data2
Number of sire-herd-year-season classes	16,891
Minimum EDC ^1^ per class	1.60
Average EDC ^1^ per class	2.9
Maximum EDC ^1^ per class	29.9
Number of herd-year-season classes	3316
Minimum frequency per hys class	3
Average frequency per hys class	5.1
Maximum frequency per hys class	29
Numbers of sires	483
Minimum frequency per sire	3
Average frequency per sire	35.0
Maximum frequency per sire	781
Number of herds	406
Minimum frequency of HYS per herd	3
Average frequency of HYS per herd	8.2
Maximum frequency of HYS per herd	25

^1^ EDC: effective daughter contribution.

**Table 3 animals-12-01189-t003:** List and definition of the environmental covariates as they were used in the 8 models.

Model	Definition of Environmental Covariates
BASE	Uncorrelated HYS classes.
AWN	Covariance based on the 1060 WVN ^1^.
PC2	Covariance based on the first 2 principal components of the 1060 WVN.
PC10	Covariance based on the first 10 principal components of the 1060 WVN.
PC20	Covariance based on the first 20 principal components of the 1060 WVN.
PC30	Covariance based on the first 30 principal components of the 1060 WVN.
PC40	Covariance based on the first 40 principal components of the 1060 WVN.
PROD	Covariance based on MY and SCS.

^1^ WVN: Fourier-transformed infrared wavenumber.

## Data Availability

Data are property of Lactanet Canada, The Semex Alliance and the individual dairy producers. Data may available under a research agreement.

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
