# Peer review of "Use of Milk Infrared Spectral Data as Environmental Covariates in Genomic Prediction Models for Production Traits in Canadian Holstein"

_animals, 2022, doi:10.3390/ani12091189_

Round 1
Reviewer 1 Report
This is a novel study about incorporating milk infrared spectral data in genomic evaluations. The topic is very interesting; however, the manuscript is too long, and the work is not presented well. As a reader, I was very interested in seeing how spectral data was modelled and how it improved genetic evaluations. The authors provide so much other information and details, probably to keep the best for the last, that the reader gets out of patience to hear about what matters the most. The trait/species should be indicated in the title, and I am not convinced that spectral data are environmental covariates. Environment influences those, but there are genetic components in them.
The authors explain the procedure in Materials and Methods, but they do not mention why they are doing things that way. In fact, there are several models, BLUE, BLUP, GBLUP, with/without genomic information, with/without GxE, and the model for variance components estimation is different from the genomic evaluation model. It is difficult for the reader to follow what is happening and what the authors are trying to achieve.
Another important thing is that the methodology does not resemble a real genomic evaluation done in practice. Estimating HYS BLUEs is not a common practice. Similarly, de-regressed proofs or DYD (corrected for HYS) are used rather than DYD within HYS (hysDYD). Again, the logic is not explained, and there are many scenarios and information that the real topic gets lost in the way.
Minor comments:
Because there are no line numbers, it is almost impossible to comment.
Simple summary and Introduction: I have not encountered such a definition of genomic selection. What is the role of genomics here?
Page 1, last two lines: “environmental effects could depend on the genotypes”, This is not true, and it does not mean GxE.
Figure 1: This figure might be unnecessary.
Please clearly explain the problem and how you want to address it.
Page 4: Indicate HYS and herd rather than HYS identifier and herd identifier.
Author Response
This is a novel study about incorporating milk infrared spectral data in genomic evaluations. The topic is very interesting; however, the manuscript is too long, and the work is not presented well. As a reader, I was very interested in seeing how spectral data was modelled and how it improved genetic evaluations. The authors provide so much other information and details, probably to keep the best for the last, that the reader gets out of patience to hear about what matters the most. The trait/species should be indicated in the title, and I am not convinced that spectral data are environmental covariates. Environment influences those, but there are genetic components in them.
The authors explain the procedure in Materials and Methods, but they do not mention why they are doing things that way. In fact, there are several models, BLUE, BLUP, GBLUP, with/without genomic information, with/without GxE, and the model for variance components estimation is different from the genomic evaluation model. It is difficult for the reader to follow what is happening and what the authors are trying to achieve.
Another important thing is that the methodology does not resemble a real genomic evaluation done in practice. Estimating HYS BLUEs is not a common practice. Similarly, de-regressed proofs or DYD (corrected for HYS) are used rather than DYD within HYS (hysDYD). Again, the logic is not explained, and there are many scenarios and information that the real topic gets lost in the way.
AU: We appreciate the time spent by the Reviewer on our manuscript. We have modified the title to include the species and traits under study. We’ve added a sentence at the end of the abstract to address the main point of the manuscript, unfortunately there is not much room for adding arguments.
The spectral data is usually not considered as for its environmental component, but the message that goes with our manuscript is just about it. The spectral information contains environmental information that can be used, after the proper statistical treatment.
The manuscript is unfortunately long, but the description of the statistical treatment and modeling for the incorporation of spectral information is long as well. As for now, this is the statistical procedure that we can propose and test. Hopefully, further research will deliver a faster, more efficient procedure.
The use of hysDYD was chosen following up on previous work on ours (Tiezzi et al., 2017). This was dictated by the use of the software BGLR and Kernel regression. If raw phenotypic data was used, we would have had to fit Kernels of number of rows (and columns) equal to a few hundreds thousands, which was unfeasible. Therefore, we had to condense information in the hysDYD. Future studies will have to explore the use of different software to allow the inclusion of raw phenotypic information rather than the hysDYD.
Minor comments:
Because there are no line numbers, it is almost impossible to comment.
AU: We apologize for the inconvenience. Line numbers were added.
Simple summary and Introduction: I have not encountered such a definition of genomic selection. What is the role of genomics here?
AU: The definition has been improved in both the Simple summary and Introduction. Lines 15 and 38-39.
Page 1, last two lines: “environmental effects could depend on the genotypes”, This is not true, and it does not mean GxE.
AU: This was partially rephrased. We meant that GxE, while it implies different reaction to environmental changes depending on the genotype, it also implies that the environmental changes will exert different effects on each genotype. Line 50.
Figure 1: This figure might be unnecessary.
Please clearly explain the problem and how you want to address it.
AU: We prefer to keep this picture in because it’s a representation of the different scenarios that could challenge a model that include genetic but also environmental effects. We have improved the figure caption and partially rephrased the sentences in lines 81 to 86. In summary, our message is that the prediction under conditions as in ‘D’ is the most challenging and could make beneficial use of the spectral information.
Page 4: Indicate HYS and herd rather than HYS identifier and herd identifier.
AU: Fixed. Lines 140-141.

Reviewer 2 Report
General comments
This manuscript represents an innovative approach to genomic prediction in a new environmental condition defined by means of milk spectra to be used as an environmental covariate. The possibility of using predictions for new animals (i.e., sires) in new environmental conditions (possible actual situation because of rapid changes in management or climate) is exploited and validated through a very robust dataset based on milk yield and somatic cell score. Similar studies could be considered new for animals, although to my knowledge there is some literature in plants that could be used further by authors as reference (please see Saint Pierre et al., 2016. Scientific Reports 6:27312 DOI: 10.1038/srep2731). The manuscript is generally well written, results and discussion are very sound with respect to the aim of the study, and so it is for conclusions. My only concerns are on the quality of figures, particularly figure 1, where some dashed lines are stated but not available or visible. I would suggest the authors check why Figure 1 is not totally visible as indicated. Apart from this and only some small changes suggested below as specific points, I do not see any further concern. Therefore, in my opinion, the manuscript should be worth publication in Animals journals.
Specific points
P 7 after the model at the bottom part of the page: geij rather than ghij.
P 10-second line under sub-section 3.3: Figure 3 and 4 rather than Figure 4 and 5.
Author Response
General comments
This manuscript represents an innovative approach to genomic prediction in a new environmental condition defined by means of milk spectra to be used as an environmental covariate. The possibility of using predictions for new animals (i.e., sires) in new environmental conditions (possible actual situation because of rapid changes in management or climate) is exploited and validated through a very robust dataset based on milk yield and somatic cell score. Similar studies could be considered new for animals, although to my knowledge there is some literature in plants that could be used further by authors as reference (please see Saint Pierre et al., 2016. Scientific Reports 6:27312 DOI: 10.1038/srep2731). The manuscript is generally well written, results and discussion are very sound with respect to the aim of the study, and so it is for conclusions. My only concerns are on the quality of figures, particularly figure 1, where some dashed lines are stated but not available or visible. I would suggest the authors check why Figure 1 is not totally visible as indicated. Apart from this and only some small changes suggested below as specific points, I do not see any further concern. Therefore, in my opinion, the manuscript should be worth publication in Animals journals.
AU: We appreciate the time taken by the Reviewer for our manuscript. We have uploaded a new Figure 1 file that is now four times the number of pixels of the old one, hopefully this one will read better. Unfortunately, we could not find the manuscript the Reviewer was pointing at.
Specific points
P 7 after the model at the bottom part of the page: geij rather than ghij.
AU: Fixed. Thank you. Line 253.
P 10-second line under sub-section 3.3: Figure 3 and 4 rather than Figure 4 and 5.
AU: Fixed. Thank you. Line 353.

Reviewer 3 Report
This manuscript is very novel, describing a method of combining genetic and environmental effects to predict animal performance in known, and in new environments. It has been long known that accurate genomic predictions may not be sufficient to accurately predict animals’ future performance. This manuscript makes an attempt to fill that gap.
Because the manuscript describes a novel methodology and is very technically oriented, it is very important that the authors clearly describe the data used, methodology, and the results. While the data used is fairly well described, I had some trouble understanding the methodology at the first reading. I think this section needs very clear description of the two step process: the first step, aimed basically at estimating effects of HYS for each variable involved, and the second step aimed at estimation of animal (or sire x HYS) effects. Having appropriate headers is necessary.
The cross-validation procedure also needs more details and explanations, as it is different from “classic” validation scheme, where genomic predictions obtained using only genotypes are compared with either adjusted phenotypes or “full” predictions based on phenotypes and genotypes. Considering that the authors use pseudo-phenotypes defined as “within-HYS” DYD for each sire, it was not clear to me what combinations of sires and HYS were included in the validation scenarios B, C, and D.
The Results section is very confusing. There were two things I could not understand:
- In Figure 2, the WVNs are labeled from 1000 to 5000 and the comments regarding this figure also talk about WVN higher than the number of WVNs mentioned before, which was 1060. If these 1060 WVNs had labels different than 1-1060, this must be mentioned somewhere. But if these values are not important for this study, just label them 1-1060. Otherwise the figure and the accompanying explanations are very confusing.
- Figures 3 and 4 (incorrectly stated as 4 and 5 in Section 3.3), top panel: Why there are two bars for all models except Base? Does one bar represent the model with GxE and the other without? So, does it mean that for the Base model there was no option with GxE (and if so, why not – the matrix of environmental effects could be just I). Better labeling of the graphs would be helpful, as some explanations in the text.
Other comments:
Figure 1: Please provide some explanations to what A, B, C, D mean, either in the caption or on the labels inside the figure. Explanation that is given 6 pages later is not very helpful.
3rd paragraph on page 5: To my knowledge, all gibbsf90 programs perform Gibbs sampling (non-parametric procedure) and do not require providing any prior (except starting values). If you used a version that allows that, please indicate which version.
Section 2.3.6: What is M0?
P.13: Why is PROD model unrealistic? Because you are fitting HYS twice?
Author Response
This manuscript is very novel, describing a method of combining genetic and environmental effects to predict animal performance in known, and in new environments. It has been long known that accurate genomic predictions may not be sufficient to accurately predict animals’ future performance. This manuscript makes an attempt to fill that gap.
AU: We appreciate the time taken by the reviewer for our manuscript.
Because the manuscript describes a novel methodology and is very technically oriented, it is very important that the authors clearly describe the data used, methodology, and the results. While the data used is fairly well described, I had some trouble understanding the methodology at the first reading. I think this section needs very clear description of the two step process: the first step, aimed basically at estimating effects of HYS for each variable involved, and the second step aimed at estimation of animal (or sire x HYS) effects. Having appropriate headers is necessary.
AU: Thanks for this comment. We added “Step 1” to “Step 4” to the headers of sections 2.3.2 to 2.3.5. Also, in section 2.3.1 we’ve added, between parentheses, the reference to each step.
The cross-validation procedure also needs more details and explanations, as it is different from “classic” validation scheme, where genomic predictions obtained using only genotypes are compared with either adjusted phenotypes or “full” predictions based on phenotypes and genotypes. Considering that the authors use pseudo-phenotypes defined as “within-HYS” DYD for each sire, it was not clear to me what combinations of sires and HYS were included in the validation scenarios B, C, and D.
AU: We appreciated this comment. We have re-written the most of section 2.3.5. We hope it’s cleared now.
The Results section is very confusing. There were two things I could not understand:
- In Figure 2, the WVNs are labeled from 1000 to 5000 and the comments regarding this figure also talk about WVN higher than the number of WVNs mentioned before, which was 1060. If these 1060 WVNs had labels different than 1-1060, this must be mentioned somewhere. But if these values are not important for this study, just label them 1-1060. Otherwise the figure and the accompanying explanations are very confusing.
AU: The labeling of the wavenumbers is not of particular relevance to this study, but we wanted to maintain consistency with a previous study that employed a subset of the same dataset (Rovere et al., 2019). That’s why we used the same term and labeling. However, this could in fact be misleading. We have then added an explanatory sentence in section 2.1 (lines 138-139) and to the caption of figure 2. We hope this can suffice.
- Figures 3 and 4 (incorrectly stated as 4 and 5 in Section 3.3), top panel: Why there are two bars for all models except Base? Does one bar represent the model with GxE and the other without? So, does it mean that for the Base model there was no option with GxE (and if so, why not – the matrix of environmental effects could be just I). Better labeling of the graphs would be helpful, as some explanations in the text.
AU: We fixed the error in section 3.3, thank you. Also, we’ve added explanatory sentences to the captions of figures 3 and 4 as well as to section 3.3.
Other comments:
Figure 1: Please provide some explanations to what A, B, C, D mean, either in the caption or on the labels inside the figure. Explanation that is given 6 pages later is not very helpful.
AU: We’ve added a short description of each section to the figure 1 caption.
3rd paragraph on page 5: To my knowledge, all gibbsf90 programs perform Gibbs sampling (non-parametric procedure) and do not require providing any prior (except starting values). If you used a version that allows that, please indicate which version.
AU: We used version 1.86 of the gibbs2f90. To the best of our knowledge, all version can be specified a prior for the variance components by using option “OPTION prior x y” (http://nce.ads.uga.edu/wiki/doku.php?id=readme.gibbs2). If not specified, the prior will be a bounded uniform distribution, as we used.
Section 2.3.6: What is M0?
AU: That was a mistake, we meant model ‘Base’. It is now fixed, see line 288.
P.13: Why is PROD model unrealistic? Because you are fitting HYS twice?
AU: We expanded the explanation in section 4.4, lines 471-477. In practical words, using the covariates in model PROD means obtaining HYS solutions for the ‘new’ environments. In order to obtain those, phenotypes need to be measured in such environments, which means that prediction wouldn’t be needed, as phenotypes are available. Thus, we used model Base just as a reference.

Round 2
Reviewer 1 Report
Title: Please delete “as environmental”. Infrared measurements are traits. Those are not environment variables because “the message that goes with our manuscript is just about it”.
L15: Genomic selection is not about “unobserved genotypes”! Please correct this definition.
L16: “these models”. Which genomic selection models are you referring to?
L16: Change “seldom” to “seldomly”.
L25: Delete “environmental”.
L36: “calibration equations” It is unclear what it is. It is better to keep the Abstract short and keep these details for later.
L43-43: This is a completely wrong definition of genomic selection.
L45: Change “seldom” to “seldomly”.
L45-50: Delete from “Under the assumption” to “In addition, these”. These lines do not give any important information rather than we should consider environmental effects in the model.
L51: Change “the different” to “different”.
L65: Change “could be considered the ‘known’ or ‘proven’ ones” to “are considered known or proven”.
L66: Change “Herds 1 to 4 could be considered as ‘known’ environments” to “Herds 1 to 4 are considered as known environments”.
L74: Similar to L65.
L75: Similar to L66.
L83: “we should keep in mind that herd management and environmental conditions are in constant change”. If it is only about the environment, why only D and not C?
L85-86: “to characterize the environments with descriptors in the same way that we characterize the genotypes with the genomic markers.” Very ambiguous! What does this really mean?
L92-98: First, this belongs to Methods, not Introduction. Second, you provide no reason why BLUE should be used.
L92-93: Change “Best Linear Unbiased Estimates (BLUE)” to “Best Linear Unbiased Estimates (BLUE) of contemporary groups”.
L92: Change “is run” to “is to run”.
L98: Change “in the new environmental classes” to “in new environmental classes”.
L100-101: “There is the need to train on the available data to conceptualize on the different use of those data in genomic prediction models.” Very ambiguous! What does this really mean?
The grammar and scientific writing needs considerable improvement. I stop making such comments for the rest of the manuscript.
The information given in the manuscript is not concise, and there are many ambiguous sentences in several places. The readability of the manuscript is not good. It needs to be concise, clear, and to-the-point. The idea of the study is brilliant. However, the authors fail to elaborate on the choice of approaches in Methods. The BLUE estimates can be considerably imbalanced and inaccurate, and EDCs cannot solve their problem. EDCs are good weighting factor for DYD and de-regressed proofs. There are several ways of calculating EDC. Only in Fikse and Banos (2001) there are 7 ways for it. The authors do not explain how they calculated EDC.
Reviewer 3 Report
The authors improved the manuscript significantly by adding details and clarifications and improving the structure of the paper. The manuscript is now easier to read and understand. I also very much appreciate adding line numbers for easier reference.
A couple of important questions still remain:
- Line 186-189: Please explain in detail how you used priors in the Gibbs2f90 programs. You mentioned OPTION prior x y that “a degree of belief” in the reply to reviewers only. Please rewrite this sentence to enable readers to fully understand what you did and how; provide as much details as possible for those who may want to do similar analyses.
- Line 269-270: “The models were fit on the training set in order to obtain solutions and predictions were obtained for the validation set”. Does it mean that for validation, phenotypes of animals from selected sires and herds were removed from the data and the solutions for those sires/herds were obtained from correlations (using G matrix for sires and whatever (co)variance matrix was used for HYS effects in different models)? Or did you use two-step approach? Please explain and provide more details.
Other minor comments:
Line 58: For example, climate change could lead …
Line 85: Therefore, there is a need …
Line 117: … have been used to successfully …
Line 127: … of the known environmental range …
Line 147: … performed by removing …
Line 158: Out of 1,992 genotyped bulls, how many were included in the data as sires of cows with phenotypes?
Line 253: … as defined in Equation [3] …
Line 509: … could be particularly …
